# Role of Blockchain Technology in Combating COVID-19 Crisis

**Zaina Alsaed [1], Raghad Khweiled [1], Mousab Hamad [1], Eman Daraghmi [1], Omar Cheikhrouhou [2,3], Wajdi Alhakami [4] and Habib Hamam [5,6,7,*]**

[1] Faculty of Graduate Studies, Palestine Technical University—Kadoorie, Jaffa Street, Tulkarm 9993400, Palestine; z.i.saed@students.ptuk.edu.ps (Z.A.); r.f.khweiled@students.ptuk.edu.ps (R.K.); m.a.hamad4@students.ptuk.edu.ps (M.H.); e.daraghmi@ptuk.edu.ps (E.D.)

[2] CES Laboratory, National School of Engineers of Sfax, University of Sfax, Sfax 3038, Tunisia; omar.cheikhrouhou@isetsf.rnu.tn

[3] Higher Institute of Computer Science of Mahdia, University of Monastir, Mahdia 5111, Tunisia

[4] Department of Information Technology, College of Computers and Information Technology, Taif University, Taif 26311, Saudi Arabia; whakami@tu.edu.sa

[5] Faculty of Engineering, Université de Moncton, Moncton, NB E1A3E9, Canada

[6] School of Electrical Engineering, University of Johannesburg, Johannesburg 2006, South Africa

[7] Spectrum of Knowledge Production & Skills Development, Sfax 3027, Tunisia

**\*** Correspondence: Habib.Hamam@umoncton.ca

**Abstract:** The COVID-19 pandemic has negatively affected aspects of human life and various sectors, especially the health sector. These conditions led to the creation of new patterns of life that people have had to deal with to reduce the spread of the epidemic by committing to social distancing, among others. Therefore, governments and technological organizations had to take advantage of technological developments in the current era to overcome these challenges that were created by these conditions. In this paper, we will discuss the role of the blockchain in combating the COVID-19 crisis. Then we will review the recently recorded blockchain-based research proposals to control the COVID-19 pandemic. Finally, we will highlight the challenges of using blockchain to combat the COVID-19 pandemic and find solutions to mitigate these challenges.

**Keywords:** Coronavirus; COVID-19; blockchain; healthcare; security; epidemic

## 1. Introduction

In the first quarter of 2020, the world started a new form of life due to the COVID-19 pandemic, with the increasing number of affected to nearly 166 million and over three million deaths in May 2021 [1]. Countries had to enforce bans and segregate of people to reduce infection, save lives, and reduce its spread [2–5]. Therefore, significant challenges emerged: governments, technology giants, and regulatory bodies had to find urgent solutions.

These challenges included continuing the provision of essential government services, as it was difficult to maintain social distancing and provide all government offices' services [3]. One of the biggest challenges that countries dealt with is combating misinformation about COVID-19 and preventing its spread. For example, misinformation on vaccination turnout was studied on 8001 respondents (4000 from UK and 4001 from the USA). The study's result was that the number of people who had the intention to receive the vaccine decreased to 6.2% after hearing the vaccines' misinformation [4]. Also, among the challenges was how to deal with the new learning mechanism, which was the shift to e-learning. Moreover, the health sector has been dramatically affected by the COVID-19 crisis, so it had to deal with complex challenges, such as the constant supply of medicines and health equipment [3], and providing care to patients (not just COVID-19 patients).

Research by Shah et al. proposed a secure blockchain network with compatibility to manage and store Electronic Health Records (EHR) of all patients and ease accessing historic and real-time patient data while reducing the cost of data accommodation [6]. But there is no specific explanation and description of specific incentive mechanisms in this model. Furthermore, Kassab et al. in [7] analyzed and summarized the existing advantages and challenges of assimilating blockchain in the healthcare domain. Moreover, they explained how this technology conveyed a framework that was based on smart contracts, which can find solutions for the healthcare domain for all interrelated actors.

Therefore, most countries and health sectors have used data tracking applications that are based on Bluetooth proximity tracing or geolocation tracking functionality [5,8]. It provides many advantages such as tracking COVID-19 cases, knowing their locations, sharing data, and tracking patients' status by remote medical staff [9]. But these apps lack confidentiality and integrity, making it easier to hack and obtain patient information. Thus, governments and technological organizations should take advantage of technological developments in the current era to face these challenges. For example, the use of artificial intelligence (AI), machine learning, the Internet of things (IoT), blockchain, and 3D printing, which can be used to create strategies for managing the epidemic [3]. For example, blockchain-based data-tracking applications can replace applications that rely on Bluetooth affinity as the blockchain can maintain confidentiality and integrity.

In this paper, we focus on the role of the blockchain in managing and combating the COVID-19 crisis. Blockchain technology has features that enable it to revolutionize various sectors [6], including the health sector, as it is considered one of the most affected sectors by the COVID-19 crisis. Furthermore, according to the European Parliamentary Research Service (EPRS), blockchain is a critical technology to fight COVID-19 [7]. Blockchain can be used to increase the security of health data and maintain patients' privacy.

Blockchain is a decentralized technology with unique features such as impenetrable data infrastructure, confidentiality, and built-in crypto security software [10]. It is a network of peer-to-peer (P2P) computing nodes that collectively validate transactions within the network [3]. The decentralized blockchain platform is tamper-resistant due to the cryptographic infrastructure that is used to authenticate network users [10]. Possible employment of blockchain technology to fight COVID-19 includes observing infection transmission through reliable and immutable ledgers in a decentralized structure [11].

The elements of originality of the present article may be summarized as follows:

- After reviewing the potential of the blockchain-based methods to face the COVID-19 virus challenges, we demonstrated the possibility and interest of implementing blockchain technology in solving healthcare contingency.
- We showed that blockchain technology could offer various services that are associated with data privacy support.
- We identified the limitations of the successful employment of blockchain technology in healthcare during the COVID-19 era.

The remainder of the paper is organized as follows. Section 2 provides the origins and components of the blockchain and a brief background on their use history. Section 3 will discuss the research methodology and strategy. Section 4 will discuss the results of the research methodology and the applications of blockchain technology to fight against the COVID-19 crisis. Section 5 will review recently recorded blockchain-based research proposals for controlling the COVID-19 pandemic. Section 6 highlights the challenges that are associated with blockchain use in the context of the COVID-19 pandemic and provides solutions to mitigate these challenges. And finally, Section 7 presents future directions as well as concluding remarks.

## 2. Background Work and Literature Review

### 2.1. Blockchain Technology: An Overview

Let us present the most recent works that are related to blockchain that have been applied to healthcare to identify the limitations of using this technology.

Blockchain started was invented in Japan by Satoshi Nakamoto as a form of cryptocurrency system with the first type called Bitcoin [12,13]. Monrat et al. in [13] defined blockchain as "The underlying technology of several digital cryptocurrencies". A blockchain is a new form of data structure. The information is stored in a digital ledger and signed digitally by building blocks within a decentralized and distributed network. Monrat et al. explained the categorization of blockchain and the architecture, and they furnished a study based on a comparison of the tradeoff of blockchain. In addition, they compared different consensus mechanisms and discussed challenges [13]. Over time, blockchain technology has been used in finance, business, and healthcare. Abu-Elezz et al. in [12] performed a dedicated review about how blockchain technology supports the healthcare field and its menace [12]. They aimed to investigate and categorize the advantages and drawbacks of using blockchain technology in a healthcare system. They used many databases such as ScienceDirect, IEEE, Springer, PubMed, and CINAHIL [11]. For example, they categorized the threats as technical or technological threats, social threats, and organizational threats. However, the benefits can be classified into more than two types. Fusco et al. aimed to validate blockchain in healthcare and suggest a traceroute for a COVID19-safe clinical practice [14]. Liao identified various social marketing success criteria that aimed to improve health promotion performance by using an optimization method. He opted for the Fuzzy decision-making trial and evaluation laboratory (F-DEMATEL) [15]. A records collection from medical examination present the limitations that was emphasized by Shen et al. [16], who proposed an efficient data-sharing scheme called MedChain. The scheme combines blockchain, digest chain, and structured P2P network techniques with the aim of overcoming issues that are related to records collection from medical examination.

Blockchain features such as immutability, decentralization, transparency, and traceability can also be used to facilitate healthcare-related procedures [12,17]. Decentralization can be defined as the process of transferring control from a centralized entity to a distributed network [6,12,17]. Immutability is the stability of blockchain entries which means that the data will never be altered. Meanwhile, traceability is the process of tracing the data for some duration of time. In contrast, transparency is the clarity of the data which means all users in the network can see the information that is stored in the blockchain transaction [6,7,12].

As described in [12,13,17], the four elements constituting the block are information, a hash of the former block, timestamp, and the hash that represents the identification number of the current block. The former block and the new block are connected together [13]. Mainly, there are three types of blockchain technology, namely public (permissionless), private (permission), and hybrid blockchains [6,12]. As its name indicates, a public blockchain is structured to be open source and allows anyone to be involved in it. One famous example of this type is the Proof of Work (PoW) algorithm [13].

According to the dictionary, blockchain is "a chain of blocks connected by a link". A block structure is the collection of valid transactions. The transaction can be started by any node within the network and then broadcast to all nodes. Then, there is a validation process that can be done using old transactions by implementing cryptography algorithms. When these two main processes are completed, the transactions will be added to the existing blockchain. The number of transactions inside each block is not constant. Back in the days when Satoshi came up with Bitcoin, the block had approximately 1 MB. In March 2018, it grew up to 8 MB. The more transactions that are processed, the larger block that is needed, but the faster processing that is obtained [15]. Figure 1 shows how blocks are connected in the blockchain network. The link between the blocks is the reference, which

it is called the hash in binary context, and it belongs to the previous block called the parent block. The genesis block is the first one that has no parent block.

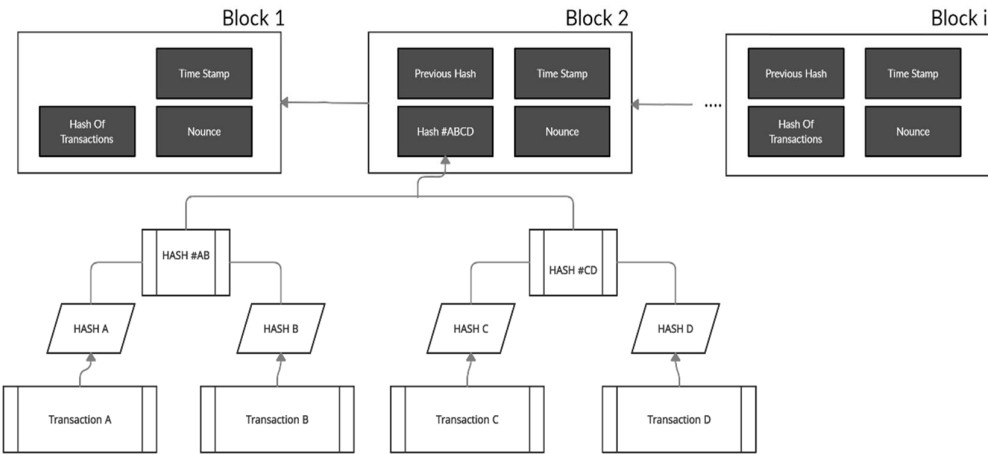

**Figure 1.** Block Connectivity.

Furthermore, all transactions are visible to all public blockchain participants that have the needed authorization. Thus, that implies no control options from any user or entity [6,12]. On the other hand, in the private blockchain system, whenever anyone needs to involve within the blockchain, he/she should request permission to enter. The transactions within this system are available upon access by authorized participants only. While the hybrid blockchains have got changeable choices, certain data can be kept private or available for the public, and in some research, they call it Consortium [6,12].

Xiao et al. in [18] presented a comprehensive review and analysis on the state-of-the-art blockchain consensus protocols. They defined the consensus protocol as the key technology underlying the blockchain's security and performance. Many consensus techniques, ranging from incremental enhancements to the Nakamoto consensus protocol to creative alternative consensus mechanisms, have been proposed to improve the blockchain network's performance or meet certain special application demands. For example, blockchain researchers are interested in innovative block proposal techniques such as proof of stake (PoS), proof of authority (PoA), and proof of elapsed time (PoET), which are not computationally demanding to mine, hence lowering energy use [18].

Zheng et al. in [19] undertook a thorough examination of blockchain technologies First, they present an introduction of blockchain architecture before comparing various common consensus techniques that are utilized in different blockchains. In addition, technological hurdles and recent improvements are briefly discussed.

### 2.2. Block Structure

A block has two main components, the body and the header [14]. The header consists of metadata as parent block hash, block version, timestamp, nBits, Merkle tree root hash, and nonce [16,20]. The transaction counter and transactions are the foundations of the block body. The transactions describe the list of recorded transactions within the block, and the counter represents the number of transactions that rest to follow [20].

Blockchain uses many algorithms within its system. For validation and authentication of the transaction, it uses asymmetric cryptography encryption. And to assure the trustworthiness of the environment, it uses the digital signatures. The digital signature is used by signing of the transaction by the user who initiated it and for the verification by the minors. Each node in the network owns a private key and public key pair for these two methods [19,20]. Figure 2 shows the block structure [21].

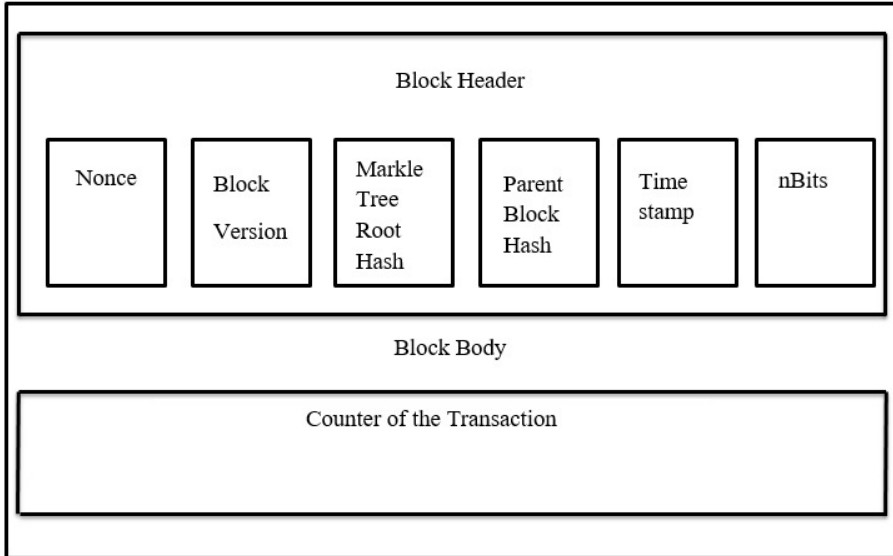

**Figure 2.** Block structure.

*2.3. Nodes Network*

The blockchain system's information is immutable when appended to other nodes. This process guarantees the distribution besides decentralizing. Consistency can also be achieved within the nodes by storing the local copy in the blockchain system and updating every timestamp. There is no trust between nodes, which can eliminate the problem of the single point of failure. Every node is responsible for executing transactions in a collective way [17,22]. The node can also perform multiple processes simultaneously, like validation, transaction, and starting mining [12,20]. Figure 3 below shows the node connectivity in a blockchain network [21].

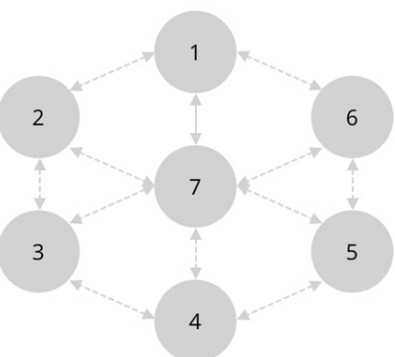

**Figure 3.** Blockchain Nodes Connectivity.

To explain how the nodes work, assume that the user starts a transaction on the blockchain network, then a new block is made. After that, this block is copied and this copy is transmitted by broadcast method to all nodes within the network but they don't add this block to their copy of the blockchain until they verify there is no manipulation in it [12,13,17].

As mentioned in [20,23], blockchain uses P2P (peer-to-peer) in a decentralized way. It uses three main algorithms and functions: hash function, digital signature, and public-private keys. As shown in Figure 1, the hash function connects the blockchain system

block to ensure that no manipulation will happen, and for an identity check, the digital signature is used. In another way, it prevents non-repudiation in the system. Researchers also state [23] that blockchain architecture can be permitted or permissionless [22], and the application is the one that decides it.

One of the biggest challenges of blockchain is scalability [24]. The authors in [25] propose a connector software, a blockchain system that is used to improve the system attributes' performance and quality. The smart contract was explained as a computational model within public blockchain in [26–28]. In [28], the authors demonstrate a blockchain tutorial and how it functions.

Framing the problem slightly differently, in [29], the trust chain can make transactions without central control. Those transactions can be used to build trustworthy between strangers too. Researchers in [29] stated that (IIS) scheme is used between the dealer and owner to confirm a new interactive incontestable signature transaction [30].

### 2.4. Smart Contracts

Smart contracts are not new technology, Nick Szabo, an American computer scientist who invented a virtual currency called "Bit Gold" in 1998 [31], defined smart contracts as computerized transaction protocols that execute the terms of a contract [32] Implementation of smart contracts was delayed during the lockdown worldwide for converting all paper-based contracts that do not have a reliable system that can handle those contracts; the paper-based system is not efficient anymore. As such, governments and financial organizations have to do something to keep the businesses running [3].

Blockchain with smart contracts is a complete solution and a better alternative to paper-based contracts. As one of the essential elements, it does not require the actual existence of the customer. In addition, it removes the delays of inherent processing and eliminates third-party intermediaries. Therefore, it is faster, lower cost, and more rapid settlement for all stakeholders [3].

### 3. Method

We conducted a scoping review of the literature on the role of blockchain technology in combating COVID-19. We followed the preferred reporting items for systematic reviews and meta-Analyses extension for scoping reviews (PRISMA-ScR) guidelines for this review [33].

### 3.1. Search Strategy

We searched nine databases on 31 July 2020: IEEE, dblp, Google Scholar, Microsoft Academic, Base search, Semantic Scholar, Arxiv, Core, and Springer link. The databases were searched using terms that were related to blockchain technology (such as blockchain and block chain) and the COVID-19 disease (such as COVID-19 and Corona pandemic). The research mechanism that was used to search in each database is explained in detail in Appendix A

We focused our research on the applied or proposed blockchain technology to help the health sector in general and fight the COVID-19 pandemic, such as contact tracing, sharing information with healthcare collaborators, and immunity and vaccination certificates. Furthermore, we focused on finding a solution to the challenges that governments and organizations faced due to the COVID-19 pandemic, such as the continuation of education and community services by government departments while maintaining social distancing.

Our research was limited to studies that were written in English language. The focus was on the role of blockchain technology in finding solutions to the challenges that resulted from the Corona pandemic. In addition, the focus was on research, articles, and

theses that were published after the Corona pandemic (i.e., in December 2019) and conferences that took place after the start of the pandemic. Also, no research was excluded due to the country of publication or study design.

Finally, we applied the stages of study selection, which contains two stages: examining the published papers and reading the entire papers. In the examination stage, Raghad Khweiled (RK), Zaina Al-Saed (ZA), and Mousab Hamad (MH) examined and filtered the research papers based on titles and abstracts. Then, the entire research that was selected from the previous stage by the reviewers (RK, ZA, & MH) was read independently. Details were discussed and any disagreements that occurred between the reviewers during the two stages were also addressed and resolved through consensus.

### 3.2. Data Extraction

The final step was data extraction that collected all of the relevant information that was required to review the technical characteristics and offerings of blockchain in the context of COVID-19. The data were extracted by the three reviewers (RK, ZA, & MH). The extracted data were collected and stored in a Microsoft Excel file and arranged alphabetically. These data included the characteristics of the studies, such as the authors' names, year of publication, and country. Also, the data that were related to blockchain technology included the type of blockchain, cost, and scalability (as shown in detail in Appendix B). A high agreement was reached among the reviewers (0.90) and solved any disagreement through consensus.

## 4. Results

### 4.1. Characteristics of Included Research

The first relevant paper discussing the use of blockchain technologies to combat COVID-19 was published in March 2020. The majority of the reviewed papers were published in May and September. The reviewed studies were conducted in nine different regions (Table 1). The features for each study are demonstrated in Appendix C.

**Table 1.** Characteristics of the included studies.

| Characteristics | Number of Studies |
| --- | --- |
| Submission month | April: 2 May: 3 July: 1 August: 1 September: 3 October: 1 |
| Country of publication | India: 2 UAE: 2 China: 1 Australia: 1 Egypt: 1 KSA: 1 Colombia: 1 UK: 1 Italy: 1 |

Appendix C shows the features of blockchain that was stated by the surveyed papers for justifying COVID-19 consequences, which are explained in the following subsections.

### 4.2. Blockchain and Healthcare

Authors in [30] explained how the distributed ledger technology could fit entirely in health services. According to [34], reports from the health research funding organization revealed that 10% to 30% of the drugs that were sold in developing countries involved forgeries. As such, a blockchain can be a perfect solution for the traceability of drugs and patient data management. WHO estimates that 16% of drug forgeries have the wrong ingredients, while 17% contain an imprecise level of the essential components. From an economic point of view, drug forgeries can cause a loss of 10.2 billion euros annually to European pharmaceutical organizations [35]. Blockchain can be a solution to address this issue because all the transactions that are added to the distributed ledger are immutable and digitally timestamped.

According to [36], the primary concern to address is data integrity within the healthcare industry. As known, each person has unique physical variability. Therefore, appropriate treatment requires a strong, proven, and successful historical medical record

of the patient. However, medical data are sensitive and require a secured sharing platform. Shen et al. have proposed Med Chain, which is created based on the blockchain of the permission framework type and it gives the patients complete control of their medical records [16].

### 4.3. Blockchain and COVID-19 Crisis- Previous Research

Since the very first beginning of 2020, in conjunction with the outbreak of the Coronavirus, the world has been trying many ways to obtain the best resolutions regarding the improvement and experimentation of vaccines to stop or limit the spread of disease. This is in addition to the immediate diagnosis of COVID-19 cases as the Coronavirus is exceptionally infectious. The blockchain's possible implementations in healthcare sectors diversify to provide different needs (e.g., access, security, and data sharing). Other forms of blockchain technologies are also built for performing some forms of clinical observations.

In the case of the current pandemic control, blockchain and its applications are emerging as a promising and effective way of providing a genuine, trustworthy, and reasonable-cost resolution to aid epidemic management, which could efficiently take place in offering a more progressive action against this ongoing pandemic. Blockchain is yielding good prospects to be an invaluable technology for stopping the spread of the Coronavirus. Its implementation would facilitate rapid observation explications, guarantee a direct supply chain of essential goods and contributions, and safe, protected payments.

Ting et al. [37] investigated how diverse modern technologies could help control the expansion of the COVID-19 illness. Specifically, the researchers in this paper note how blockchain technology; artificial intelligence (AI), besides big data technology; and the Internet of Things (IoT) technology can contribute to designing simulation patterns that predict the disease scope's expanse; these methods can support establishing a monitoring instrument that can help diagnose and observe the virus's range. Concerning real-life or daily use-cases, the authors said that the blockchain's adoption improves the tracking process of shipments of medicines to China's cases' houses. The core of this study is to state the possible use of cutting-edge methods to minimize the disease's infection. However, this research did not present aspects of specific technical applications or implementations.

In contrast, Torky and Hassanien [38] suggest a proposal to employ blockchain to automatically identify COVID-19 infected patients and determine the outbreak risk of the COVID-19 among the public. The writers employed the decentralization characteristic of blockchain to collect the information of the approved COVID-19-patients. The identification of positive-tested records relies on using extra methods, e.g., a disease verifier subsystem.

Furthermore, Nguyen et al. [39] recommend an advance to help forecast the extent of the COVID-19 infection and other related pandemics. They advised utilizing AI accompanying blockchain to treat a considerable size of medicinal data with a complicated design. The research illustrated a blockchain-based methodology to support the healthcare supply chain and some donation follow-up. Nevertheless, it does not give professional details about the application of this method.

Zaabar et al. proposed a blockchain-based structure to enhance the robustness of the healthcare management systems as well as to avoid noted security shortcomings in presently applied architectures towards smart healthcare systems [40].

Moreover, Bansal et al. [41] explained the use of blockchain in generating online immunity documents. The writers suggested using the immutability attribute of such technology to circumvent the extent of misinformation. The proposed clarification also tries to discuss the problem of secrecy and anonymity of the test-doers.

Yet, the researchers have not introduced an implementation outline to complete the outcomes of the scheme. Resiere et al. [42] offer a system that is based on blockchain technology to revive the healthcare sector in the Caribbean. Accordingly, the research recommended using blockchain to obtain healthcare assistance, plus joint systematic examination to combat the crisis of COVID-19.

Frikha et al. have proposed a blockchain-ethereum-based architecture which is mainly an integrated IoT blockchain web and mobile application to store and check electronic health records (EHRs). The proposal allows the patient and the medical staff to access health information securely [43].

To conclude, Kumar et al. [44] have formed a proposal to enhance the identification of COVID-19 cases according to tomography (CT) slabs using a deep learning model. They matched their aimed method with other AI deep learning methods, e.g., AlexNet, DenseNet, VGG16, and Capsule Network. The study applies blockchain technology as a system of distributing information with particular consideration given to preserving secrecy.

Although various visions of adopting cutting-edge methods such as IoT and blockchain to improve combating Coronavirus have been investigated, the present study attempts to propose deep procedural specifications [44]; in [43], the researchers clarify their deep learning model technical employment in different parts. The explications that are mentioned above do not decrease the spread of the virus by the straightforward method of using blockchain. In most of the mentioned research, blockchain technology is introduced as a likely technology to control the infection, assist in curbing the misinformation campaigns, or guide other technical methods to suggest a whole structure. None of the studies stated or completed resolution that was based on blockchain technology, supporting the need for marking and tracing COVID-19 likely-cases that have been verified by immunity records.

### 4.4. Applications of Blockchain Technology to Fight against COVID-19

Blockchain has a significant impact in combating the crisis of COVID-19. It is one of the main high-end technologies that plays an important role in this situation. This section provides the main applications of blockchain that help the most affected fields keep them running with the least damage. As shown in Figure 4, those applications include contact tracing, sharing information with healthcare collaborators, immunity and vaccination certificate, smart contracts, supply chain management, online education and secure certification, and e-Government.



**Figure 4.** Contributions of blockchain technology during the COVID-19 pandemic.

### 4.4.1. Contact Tracing

Contact tracing is an essential element of the control of dangerous epidemic outbreaks. It intends to stop disease transfer and has played a role in combating the COVID-19 outbreak in several regions [45]. It is necessary to calculate the cross ponding physical contact of any case during its' incubation period. This can lead to a lower extent of spread

that is caused by COVID-19 [36], according to the researchers in [3], the incubation of SARS-CoV-2(COVID-19) for starting the symptoms would take almost 5.5 days, and as described in previous sections, not all infected people got appearing symptoms [3].

In response to the surprising outbreak of COVID-19, many authorities have presented noticeable trends to mobile contact tracing apps to assist the hard mission of tracking every contact of recently recognized infected people [46]. The social interaction can be determined using contact tracing. It aims to decrease the spread of COVID-19 by taking the initiative which includes giving the individuals the analysis of the virus extent. The analysis is followed by instructions that can be diagnosed, advised, and quarantined in real danger where it is needed. Smartphones and IoT can make this application more effective. Although the users' privacy policy will be violated, the authorities and the healthcare fields adopted the contact-tracing application [36]. The data that are collected will have high accuracy and reliability if the blockchain is used in high frequency [36]. It gives real-time data for the affected areas. However, the safe zones or free-virus zones are monitored and reported by blockchain [35,46,47]. Blockchain technology integrates with other technologies to collect data, such as geographical information systems (GIS) and AI [35,46].

Two systems implement the contact tracing: the mass surveillance system and the P2P-mobile application [37,47]. The mass surveillance system depends on e-Governments and IoT devices within the streets of smart cities. In contrast, the P2P-Mobile application uses services to send the surveillance system and the blockchain network the collected data; it can show users the probability of getting infected by COVID-19 [48].

### 4.4.2. Sharing Information with Healthcare Collaborators

Sharing the patient data, treatment methods that were used, prevention methods that were used by doctors and healthcare personnel, and so on are important data that must be shared among healthcare collaborators. These data play a fundamental role in COVID-19 research as shared data from health organizations and hospitals can discover new treatment methods that help reduce the number of deaths. Therefore, international bodies and organizations use HIPAA controls to maintain the data's confidentiality, especially patient information [3]. However, there are obstacles in implementing the data-sharing system more broadly.

Blockchain features such as decentralization and transparency can be used to improve the healthcare system. It can take advantage of decentralization to increase the security and privacy of patient data, increase data control, and get rid of traditional medical records, which can be stored data in a blockchain-based system, making sharing data between health organizations and hospitals much easier and faster. The transparency feature can also be used to prevent data fraud and protect patients' privacy from abuse or misuse [3].

### 4.4.3. Immunity and Vaccination Certificate

Immunizer, serology, or antibody tests are the tests that determine if someone has gained immunity to the virus (e.g., COVID-19) upon complete healing from the infection. The immunity (vaccination) certification records the viruses or diseases that he/she have been protected or vaccinated against. The certificate helps limit and restrain the scope of COVID-19 by letting the officials and authorities form plans by supporting cross-border transportation from one place to another for those who hold this certification [49]. Accordingly, the certificate's fraud security, robust privacy means, and cost-effectiveness are the governments' fundamental responsibilities to reduce frauds relative to traveling. Employing blockchain-based vaccine certification and immunity testing implements an adequate and reliable data administration policy that is straightforward to manage, non-vulnerable, and cost-effective [5].

Blockchain manipulates asymmetric encryption and decryption patterns [50], as well as uses digital signatures to preserve both tests and certificates. Additionally, the decentralization characteristic confirms the security of such data against malicious attacks or a single point of failure by enhancing users' confidence by maintaining information authenticity and protection.

The above-mentioned certifications can be confirmed in a reliable, advanced manner, and secure the secrecy of the user information. For example, at resuming marketplaces after the COVID- 19 crisis, many businesses can promote and adopt the policies to allow only those workers to turn to work who hold a legitimate electronic vaccination license (immunity passport) to get back to work. Here, blockchain deployment guarantees that only "COVID-19 clear case" employees could resume working in the company. The essential traceability and transparency hallmarks of blockchain support building the data origin of the COVID-19 lab outcomes. It can also considerably help the businesses check the PCR testing kits' validity for the COVID-19 examination. It generates an immunity passport for the user to register it on the ledger.

### 4.4.4. Supply Chain Management

The ongoing pandemic is causing massive damage to the supply chain worldwide as factories have been unable to deal with this shutdown due to the new measures, such as physical contact and social distancing. Therefore, the ban has been applied to importing and exporting the global supply chain [3]. According to [51], "94% of Fortune 1000 companies are seeing supply chain disruptions from COVID-19", "75% of companies have had negative or strongly negative impacts on their businesses", and "55% of companies plan to downgrade their growth outlooks". Medical and pharmaceutical supply chains are hard to find because of the high demand [3]. Furthermore, the supply chains of household needs have also had a high surge in demand [3].

One of the best options to consider managing the supply chain is blockchain. First, it can connect all of the stakeholders through one decentralized universal network, and by its' property transparency, it securely shows the data of the silos. As mentioned in [36], many blockchain networks in the supply chain management field were mentioned. As mentioned in the previous sections, blockchain technology in any field or application provides quick data handling, reduces processing time, and has a lower risk in its operations. One example that was applied in China is the VeChain platform, assuring the credibility and reliability of masks that were imported from China. It checks codes, materials, packages, and all associated tasks [35,51].

### 4.4.5. Online Education and Secure Certification

Since many schools and universities have closed in different countries due to the quarantine, this does not mean stopping, or having an intermission in, the education process. Education is irreplaceable and making it an online operation is a challenge itself. Moreover, there are some associated challenges such as poor cross-platform collaboration and difficulty verifying students' credentials (degree, transcripts, and other certificates) [3].

Blockchain-based online education platforms can mitigate these challenges [3]. Decentralized blockchain-based storage can provide access to authorized users while keeping the data of others secure [3]. As known as a blockchain, it gives secure issuance and sharing of verifiable educational credentials where one of the issuing authorities of an institute uploads a credential to a blockchain-based system [4]. One of the best examples is that the MOODLE platform, a global blockchain-based network, offers the mentioned advantages [3].

4.4.6. E-Government

The Corona pandemic, lockdown, and quarantines have caused the suspension of many businesses and services in the governmental institution's sector. This has prompted many governments that do not primarily depend on information and communications technology (ICT) to facilitate their services to switch and use the concept of electrical Government, also known as e-Government, to digitize all of their services [3]. These services can be public utilities (water, gas, electricity, sanitation, etc.), tax collection, salary payment, etc. [3].

Blockchain gives high efficiency to the e-Governments systems by decreasing the delays and reducing the service operations costs. In addition, it gives access to the automation feature with blockchain and the shared databases [3]. If any counterfeiting endeavor happens, it will automatically be detected. When it comes to security, blockchain has a lot of ameliorating for data confidentiality and consistency. Data integrity and immutability are some of the benefits that are provided to e-Governments involving blockchain technology [3].

Table 2 summarizes the uses of blockchain technology in fighting the COVID-19 pandemic, such as the type of blockchain that can be used in the various applications of blockchain and the transaction cost.

**Table 2.** Existing applications for blockchain [3].

| Application | Type of Blockchain | Transaction Cost | Expected Latency | Expected Scalability |
|---|---|---|---|---|
| Contact Tracing | Public | 0–0.01 USD | Within hours | 360–1440 transactions per phone per day |
| Sharing information with healthcare collaborators | Consortium | 0–0.01 USD | Within Seconds | 1000–100,000 transactions per minute |
| | Private | No Cost | | |
| Immunity and vaccinationn certificate | Consortium | No Cost | Within hours | NAN. |
| Smart contracts | Consortium | 0–0.01 USD | Within days | 10–100,000 transactions per institute per day |
| | Private | No Cost | | |
| Supply chain management | Private | No Cost | Within minutes | 100–10,000 per system per day |
| | Consortium | 0–0.01 USD | | |
| Online education and secure certification | Public | 0–0.01 USD | Within hours | 100–10,000 per municipality per day |
| | Private | No Cost | | |
| e-Government | Consortium | 0–0.01 USD | Within days | 100–10,000 per municipality per day |

**5. Blockchain-COVID 19 Ongoing Research**

This section demonstrates the modern explorations that are underway that are presently adopting blockchain technology to deal with critical emergency status due to the COVID-19 outbreak.

*5.1. Blockchain-Based P2P-Mobile Application Design*

In response to significant issues that are related COVID-19, a proposed solution is to examine the outcomes of generating a blockchain-based, decentralized peer-to-peer communication, real-time processing mobile application scheme for assisting all people in identifying the spots of infection within a crowded group of people. The new form of the suggested P2P-mobile application will support users to automatically measure the outbreak likelihood of the COVID-19 virus propagation and to aid the authorities in discovering hidden infected sufferers. The system is mainly based on four subsystems, a P2P-mobile application, a blockchain platform, infection verifier subsystem, and a mass-surveillance system [38].

### 5.2. Telemedicine and Remote Healthcare Systems

Remote healthcare is an example of prevention means that concerns the spread of predatory use of Telemedicine. Medicalchain, a platform that was implemented on an Ethereum and Hyperledger base, has been applied to perform remote assistance that is related to medical consultancy credentials. The system ensures that patient-doctor data that are processed through the consultancy process is protected and stored correctly. After filling the agreed-upon consent mold for marketplace requests, the system allows the patient to confidentially share the information with researchers as a third party according to the filled form [52].

Another proposed solution, HealPoint, suggests the most relevant doctors to a user (patient) according to specific determinants such as geographical region, knowledge, and skills. It also permits cases to take another medical opinion remotely [53].

### 5.3. Preventing Fake News Circulation Using MiPasa Platform

The COVID-19 crisis has revealed the urgent requirement for a secure, convenient, reliable, precise, and robust system that could fix non-scalable systems that already exist for fighting the COVID-19 epidemic. Data that are associated with the COVID-19 verifiability process can profoundly reshape decision-making for numerous activities. One of blockchain technology's proposed platforms is MiPasa, which presents a blockchain-based floor that blends, processes, and provides data that are linked to the COVID-19 infection growth from versatile, valid references, e.g., WHO with certified health associations and officials. It mainly assists governments in recognizing both individual faults and falsifying by allowing data specialists and public health directors to share clarifications to restrict the expansion of the epidemic. For example, by applying data analytics on confirmed blockchain-based information, MiPasa can help identify the COVID-19 cases and virus hot points in a secret, reliable, and convenient way. It is designed as a completely protected system that is executed on top of a hyperledger framework. Users and public health agents can utilize MiPasa to place the infected cases by uploading them on web-based interfaces.

To acknowledge such, it verifies it by handling data that are supported by WHO, for example, to ensure that the uploaded data meets the primary requirements. Next, if the data is confirmed, it is distributed to the state governments and health officials that are appointed by the country [54].

### 5.4. COVID-19 Testing Privately Using Epios

Many examples of prejudice, discrimination cases, insults, and abuse are recorded globally through the COVID-19 period, in which people that are infected in COVID-19 are targeted for causing pandemics and its prevalence. To dodge such discrimination, clinics or labs should maintain the secrecy of the COVID-19 diagnosed cases. Epios is proposed to assist in secret testing of people that are experiencing COVID-19 infection. It allows them to smoothly contact the labs which provide and prepare PCR examination swabs. The proposed system guarantees that the fee cannot be paid straight to the test labs. Alternatively, it demands the COVID-19 testing kit suppliers to give a ticket for each kit to people who use the system. Besides, it cryptographically preserves the token (coupon) to help the laboratories to check the fees instead of tracking the users that are buying PCR kits.

Further, it tries to execute a mobile application that facilitates acquiring and submitting such kits in both directions, to and from the PCR test providers, in an anonymous manner. The design also proposes sharing the COVID-19-relevant data, such as exclusive COVID-19 revolution statistics with the researchers, government, and authorities while maintaining the privacy of tested people [55].

*5.5. Controlling Disease from Expanding Using Anonymous Contact-Tracing-VIRI Platform*

Restriction of the spreading of transmissible infections, such as COVID-19, is one of the objectives of digital contact tracing. By identifying infection hotspots and locating infected people, user information secrecy is affected. The main technical variations of the contact tracing resolutions and the platforms, as mentioned earlier, can change the flexibility and efficiency of digital contact tracing's proposed technologies. Among these, the cross-entity VIRI's platform has proposed a universal platform to solve such research problems widely. It also guarantees to conserve the user's data privacy.

VIRI was designed to trace the scope of the disease in different regions, which can effectively handle locating the growth of COVID-19 infection at several spots. Whenever a user has had close contact with a COVID-19 positive-tested person, it informs him/her by pushing a notification. The system can warn the user regarding dangerous illnesses in the following step according to the hazard scale. For example, when a user has had direct contact with an infected case, VIRI can modify the state of a user of a "cleared case attribute" to a "likely COVID-19 case".

By open-source APIs, there is an option of easily integrating the platform with already existing enterprise technologies. Furthermore, VIRI confirms that the individual has full authority over their information, and it is saved on their device anonymously using blockchain technology by benefiting from its privacy-preserving characteristics. Such can support AI-based means to predict the COVID-19 pandemic globally [55,56].

*5.6. Self-Sovereign Identity Using COV-ID and E-Rezept Systems*

An application of remote platforms in the healthcare sector is E-Rezept, that is built on the concept of self-sovereign identification. Its main objective is enabling cases to remotely set a purchase for medications by showing their unique identifiers as evidence [57,58]. In distinction to Medicalchain and HealPoint, this platform is manageable and seamlessly combined with present healthcare systems. Another instance of blockchain-based startups is COV-ID that is based on Sovrin, a permission self-sovereign identity (SSI) network [59]. The COV-ID platform provides premia to law-abiding civilians in a liable and reliable way [60]. An application of media that is based on blockchain technology is VeChain. It observes vaccine progress and guarantees the dependability of data that are related to vaccine details [61].

## 6. Challenges of Using Blockchain during the COVID-19 Pandemic and Possible Solutions

This section will focus on the challenges that can be addressed when using blockchain technology to manage and combat the COVID-19 pandemic and find possible solutions to reduce these challenges.

*6.1. Smart Contracts Security*

Smart contracts can be used to verify COVID-19 vaccination certificates, issue immunity passports for individuals, and implement the terms that are agreed upon between companies and organizations. Also, through smart contracts, cryptocurrencies and assets such as vaccines or PPE can be tracked [61].

Although many advantages can be used for smart contracts, including ease of correction and cost efficiency, some challenges have arisen due to how blockchain technology is designed that we may face when using smart contracts in managing the COVID-19 pandemic. Among these challenges, public blockchain platforms are open source; data and transactions can be accessed from the public, exposing the system to tampering by malicious users. Also, the system that it operates on is decentralized as smart contracts can be exposed to many security threats from malicious users who use pseudonyms and control nodes for malicious purposes. For example, malicious users can manipulate smart contracts that use control nodes to extract fake vaccination certificates for people who have

not received the vaccine. In addition, the use of blockchain technology by people who are not experienced in the technology leads to flaws in the design of smart contracts [61,62].

According to [63], these tools (FSolidM, KEVM, Securify, MAIAN, and Mythril) help discover vulnerabilities and secure smart contracts against any possible attacks. Although techniques and tools are used to reduce the gaps and challenges facing the users of smart contracts, it is necessary to propose security protocols that are based on specific goals to increase security in smart contracts [61].

### 6.2. People's Privacy Preservation

Digital contact tracing is an important role that has been proposed in using blockchain to combat COVID-19 [61]. The COVID-19 tracking applications are created based on blockchain technology to ensure the social distancing that is imposed by governments and use applications to monitor COVID-19 patients and follow up their cases.

Although there are advantages to blockchain technology that is used in the COVID-19 tracking applications, challenges must be considered when using blockchain in applications. Among these challenges is preserving user data privacy and sensitive information such as national numbers and home addresses [39]. Due to public blockchain platforms (such as Bitcoin), anyone can find out users' data. Thus, people's privacy is vulnerable to abuse.

Therefore, governments should impose laws and reinforcements to curb these violations [39]. Also among the suggestions that have been made to solve these challenges was using the mixing technique or anonymous when using blockchain technology [64]. Alternatively, we can use private blockchain platforms (like Quorum) that are operated in a controlled environment to preserve user data [61].

### 6.3. Processing Data Related to COVID-19

The data relating to the COVID-19 pandemic is large and high velocity. For example, the data that are extracted from digital contact tracing requires monitoring geographical data while determining the time and updating it regularly on the blockchain. Increasing the velocity of data creates challenges for how to process it [61].

Public blockchain platforms are of limited transaction for throughput and privacy (20 transactions per second), which leads to problems when used in processing data that are related to COVID-19 [65]. In contrast, the private blockchain platforms are fast and secure (several thousand transactions per second), but they cannot be relied upon to solve Corona's data processing problem [61,66]. The transaction rate can be reduced by incorporating an additional edge or fog-based layer into the existing data pre-processing frameworks [67]. Also, communication through external channels and data compression techniques can reduce the rate of transactions, leading to solving the data processing problem [36].

### 6.4. Create an Interoperable Blockchain-Based System

When building an interoperable blockchain-based system, we can take advantage of interoperability features that allow disparate blockchain-based systems to communicate uninterruptedly with each other [68]. Moreover, it enables users to view and share information across different blockchain platforms without assistance (translation service), so supporting the interoperability of the blockchain platform can increase productivity and supply chain of PPE and vaccines [61,68,69].

The difference in software designs and the diversity in the technologies that are used for blockchain platforms have created challenges when creating an interoperable blockchain-based system for COVID-19 outbreak [5,36].

There are no solutions that have been proposed to solve these challenges that are due to the difference in the supported languages, the recommended consensus protocols, and

data protection and transactions in smart contracts [35]. Therefore, there must be more research to find a solution to this challenge.

### 6.5. Limited Adoption of Blockchain Technology

The COVID-19 epidemic has changed the lifestyle of individuals and society, forcing governments and organizations to find solutions and use new technologies to deal with the conditions that have been created by the epidemic. Therefore, blockchain technology was used for digital contact tracing. Digital contact tracing, as mentioned, is beneficial to governments or authorities in reducing the spread of the Coronavirus and applying social distancing between citizens. Blockchain technology features, such as traceability and decentralization, can also facilitate importing PPE, vaccines, and medical supplies by increasing trust between governments and manufacturers [70]. In addition, it can be used to digitize the vaccination certificate to reduce fraudulent certificates (via traceability feature) [61,71].

Although there are advantages in blockchain technology that can fight against the COVID-19 pandemic, there is a slow and limited adoption of blockchain technology. One of the main reasons for this is: (A) there are no laws to manage blockchain technology or specific plans to manage risks; (B) lack of encouragement and initiative by companies to change their traditional commercial practices; (C) lack of sufficient expertise in blockchain technology; (D) the high rate of energy consumption affects the efficiency of mining operations for a system-based blockchain, which affects its ability to adapt; and (E) the spread of miners operations around the world and the lack of privacy laws that are applicable to them have led organizations to abstain from using blockchain technology [38,61,72,73].

Therefore, there should be more research and finding laws and standards that can be used in blockchain technology to improve adaptation by organizations and companies to combat COVID-19.

## 7. Conclusions and Future Directions

### 7.1. Future Directions

Here, we present some directions and prospects for future research regarding blockchain technology utilization for upholding the combating of the COVID-19 pandemic.

#### 7.1.1. Security Concerns of Blockchain

Notwithstanding its magnificent aspects, blockchain endures security matters. Accordingly, improving blockchain by applying new security approaches should be considered. For instance, to improve the effectiveness of the mining method, a mining pool approach is offered in [74] by explaining security bottleneck issues. In this context, security-defense tools are consolidated for battling data vulnerabilities in the consensus protocols within the blockchain. A recipient-oriented-transaction proposal is presented in [75], to alleviate double-spending crimes in blockchain, working to approve the transaction before adding to the block using the terms of "stealth address and master-node". In this case, senders and recipients of the transaction follow the verifying process for each transaction in an active manner in the broadcast process [75].

#### 7.1.2. Enhancement of Blockchain-Based Systems Performance

Blockchain-based implementation should be improved to outperform from various technical aspects with the intention of executing the blockchain as a model option for crisis healthcare employment similar to the COVID-19 pandemic. For example, scalability and lightweight blockchain deployment in the context of healthcare are required to minimize data verification time and transaction latency. Consequently, applying efficient mining mechanisms along with optimizing verification operation time can be taken into account to reduce processing time to achieve high-speed data sharing, which results in robust data analysis in the context of the COVID-19. An additional probable suggestion is to establish

both local and private blockchain systems, which could reduce the size of the blockchain. Using such aids to observe the disease in a specific region for an active real-time reply.

### 7.1.3. Blending Blockchain with Additional Technologies

To obtain adequate performance in resolving issues that are associated with the COVID-19 pandemic, blockchain can be combined with other available technologies to establish a robust healthcare architecture.

For example, ingenious storage and great computation potential are the fundamental peculiarities that could be benefited from the cloud. Such features could be employed besides the blockchain to deliver a more efficient performance. Additionally, drones support augmenting the contactless observation of the epidemic. Furthermore, using drones is one of the efficient ways to offer requisite conveniences to the people under quarantine where all means of transportation are suspended.

### *7.2. Conclusions*

This study reviewed the potential methods to invest and deploy the blockchain technology characteristics to fight the COVID-19 virus. We demonstrated the suggestions of blockchain technology employment, from principally the healthcare contingency point of view, to show the essential role that blockchain can present through the COVID-19 crisis. We reviewed recent existing systems that are based on blockchain to fulfill various services that are associated with data privacy support. For instance, remote COVID-19 examination, easily performing digital contact tracing, and remote patient observation. We classified and introduced some research difficulties and challenges that limit the successful employment of blockchain technology in healthcare through the COVID-19 epidemic widespread.

Finally, we suggest some future directions to improve the adoption of the blockchain within the context of the COVID-19 epidemic. Blockchain applications could be implemented to decrease network latency with a protected framework for collecting, storing, and sending critical data. The latest incorporation of blockchain with other rising technologies, such as Big Data, AI, and Cloud computing, can adequately manipulate fatal diseases similar to COVID-19.

**Author Contributions:** Conceptualization, Z.A.; methodology, Z.A. and R.K.; validation, Z.A. and R.K.; formal analysis, Z.A.; investigation, R.K.; writing—original draft preparation, Z.A. and R.K.; writing—review and editing, R.K., M.H., W.A. and H.H.; visualization, M.H.; supervision, E.D. and O.C.; funding acquisition, E.D. and O.C. All authors have read and agreed to the published version of the manuscript.

**Funding:** 1) Taif University Researchers Supporting Project (TURSP), Taif University, Kingdom of Saudi Arabia under the grant number: TURSP-2020/107, 2) Natural Sciences and Engineering Research Council of Canada (NSERC) and 3) New Brunswick Innovation Foundation (NBIF) for the financial support of the global project.

**Institutional Review Board Statement:** Not applicable.

**Informed Consent Statement:** Not applicable.

**Data Availability Statement:** Not applicable.

**Acknowledgments:** This project was supported by Taif University Researchers Supporting Project (TURSP), Taif University, Kingdom of Saudi Arabia under the grant number: TURSP-2020/107. The authors also thank Natural Sciences and Engineering Research Council of Canada (NSERC) and New Brunswick Innovation Foundation (NBIF) for the financial support of the global project. These granting agencies did not contribute in the design of the study and collection, analysis, and interpretation of data.

**Conflicts of Interest:** This is an original manuscript. This manuscript is neither submitted nor accepted anywhere. All authors declare that we have no competing interests.

## Appendix A

**Table A1.** Database table.

| DB | Blockchain | Block Chain | COVID-19 | COVID19 | Corona Pandemic | Blockchain and COVID-19 |
|---|---|---|---|---|---|---|
| IEEE | 9128 | 3324 | 4356 | 210 | 153 | 113 |
| dblp | 9986 | 580 | 4851 | 90 | 67 | 53 |
| GOOGLE scholar | 38,900 | 36,500 | 182,000 | 47,200 | 45,000 | 14,500 |
| Microsoft Academic | 39,604 | 17,999 | 50,000 | 3673 | 1282 | 170 |
| Base search | 45,328 | 57,021 | 580,101 | | 8641 | 534 |
| Semantic scholar | 35,000 | 4,150,000 | 335,000 | 8990 | 337,000 | 370,000 |
| Arxiv | 2316 | 1171 | 4302 | 137 | 50 | 37 |
| Core | 33,224 | 6,291,202 | 181,295 | | 617,940 | 213,404 |
| springer link | 14,479 | 498,410 | 58,038 | 3418 | 6078 | 1595 |

## Appendix B

**Table A2.** Search Strategy.

| Concept | Definition |
|---|---|
| | Study Characteristics |
| Author | The first author of the study. |
| Year of publication | The year the study was published. |
| Country of publication | The country the study was published. |
| Type of publication | Type of publication is how the study was published (such as journal, conference proceedings, thesis). |
| | Intervention characteristics |
| Name of developed application or technology | Name of blockchain technology if available (e.g., Ehtraz) |
| Application of blockchain technology | Contact tracing, sharing information with healthcare collaborators, immunity and vaccination certificate, smart contracts, supply chain management, online education and secure certification, and e-Government. |
| Type of blockchain in terms of access right | Public blockchains (permissionless): a public blockchain is structured to be open source and allows anyone to be involved. Furthermore, all transactions are visible to all public blockchain participants, with authorization of these transactions. Thus, that implies no control options from any user or entity private blockchains (permission): private blockchain system, whenever anyone needs to involve within the Blockchain, they have to request permission to enter. The transactions within this system are available upon access by authorized participants only. Hybrid blockchains (Consortium): hybrid blockchains got changeable choices. Specific data can be kept private or available for the public. |
| Platform | The name of the technology that runs the Blockchain (such as Ethereum, Bitcoin, R3 Corda, MedicalChain, Apla) |
| Transaction cost | Cost of each transaction |
| Expected latency | The time it takes to process one transaction (such as within seconds, minutes, hours, days) |
| Expected scalability | Expected scalability is the ability to scale with the number of transactions, maximum capacity, or the number of transactions that the network can process per second (such as the number of transactions per time). |

## Appendix C

**Table A3.** Authors and blockchain publication.

| Author [ID] | Month | Country | Publication Type | Application | Status | Access Type | Platform | Programming Language |
|---|---|---|---|---|---|---|---|---|
| Humayun | May 2020 | KSA | Published | Blockchain-based secure framework for e-learning | Proposal | Public | - | - |
| Lopez | May 2020 | Colombia | Preprint | Blockchain-based supply chain surveillance | Proposal | Public | - | - |
| Dai | Sep 2020 | China | Journal | Blockchain-enabled Iomt | Proposal | Public | - | - |
| Torky | Apr 2020 | Egypt | Journal | Contact tracing | Proposal | Public | - | - |
| Xu | May 2020 | UK | Preprint | Contact tracing | Developed | Public | Ethereum | - |
| Bahl | Oct 2020 | India | Journal | Contact tracing | Developed | Public | - | - |
| Marbouh | July 2020 | UAE | Journal | COVID-19 transmission tracking | Proposal | Public | Ethereum | Solidity |
| Hasan | Aug 2020 | UAE | Published | Immunity passport | Developed | Public | Ethereum | - |
| Fusco | Sep 2020 | Italy | Published | Infection prediction | Developed | Public | - | - |
| Nguyen | Apr 2020 | Australia | Journal | Outbreak monitoring | Developed | Public | - | - |
| Kalla | Sep 2020 | India | Journal | Smart contracts and contact tracing | Developed | Public | - | - |

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
