# Peer review of "Role of Blockchain Technology in Combating COVID-19 Crisis"

_applsci, doi:10.3390/app112412063_

Round 1

Reviewer 1 Report

This topic of manuscript is certainly important and interesting. As it has been in the abstract, the manuscript has three aims

  • Reviewing the recently recorded blockchain-based research proposals to control the COVID-19 pandemic.
  • Highlighting the challenges of using blockchain to combat the 20COVID-19 pandemic
  • Proposing solutions to mitigate these challenges.

After reading the manuscript, I can say that these objectives have not been achieved satisfactorily because 

  • Although the method of conducting the literature review is clearly described, the results of the literature review have not presented clearly nor adequately.
  • Except sub-section 4.3, which presented the previous research relevant to the topic, I could not figure out how the remaining sub-sections contribute to achieving the first objective.
  • Moreover, the cited references need to be carefully rechecked. In sub-section 4.3, reference 36 has been cited twice in two different contexts (line 260, line 281). There is excessive use of reference 6, which can not be accessed using the provided link. It has been cited in many subsections, despite its irrelevancy to some of them.    
  • Although a number of challenges have been identified, some of them are presented as general challenges of blockchain technology and they need to be addressed in the context of COVID-19.
  • The proposed solutions of the identified challenges need to be revised as some of them are merely concise recommendation rather than practical solution.

Reviewer 2 Report

The authors consider the extremely relevant topic of the blockchain applicability  to combat the pandemic. 
However, it is not clear what new trend have they found except for the already known facts?

Key flaws:
1. Why the review does not consider such highly cited publications as
    Braithwaite, I., Callender, T., Bullock, M., & Aldridge, R. W. (2020). Automated and partly automated contact tracing: a systematic review to inform the control of COVID-19. The Lancet Digital Health, 2(11). https://doi.org/10.1016/S2589-7500(20)30184-9

    Ahmed, Nadeem & Michelin, Regio & Xue, Wanli & Ruj, Sushmita & Malaney, Robert & Kanhere, Salil & Seneviratne, Aruna & Hu, Wen & Janicke, Helge & Jha, Sanjay. (2020). A Survey of COVID-19 Contact Tracing Apps.

    Abd-alrazaq A, Alajlani M, Alhuwail D, Erbad A, Giannicchi A (2020) Blockchain technologies to mitigate covid-19 challenges: A scoping 
    review https://www.sciencedirect.com/science/article/pii/S266699002030001X

   Perhaps the proposed method for selecting sources should be adjusted, or the selection was carried out incorrectly?

2. Section 4.4.4. Smart Contracts
  In the section there are only general words about Smart Contracts, where does the specific application to combat COVID in this section? It seens this section should be part of section 2 in the general description of DLT.

3. Instead of describing in section 2 the general principles of blockchain operation, which the reader can easily find himself, see, it is necessary to substantiate - what new this review reveals in comparison with  previously done.

4. There are a number of more detailed and complete overviews for each of the issues described here (see issue 1 above). Perhaps at the end of 2021, some new moments can be noted compared to 2020?

5. In the conclusion:
Blockchain applications could be implemented so that network latency would be decreased with a protected framework for collecting, storing, and sending critical data.

Section 6.3 states that it needs to be reduced, but why? we are not dealing with financial transactions. So if the transaction processing time is 1 hour, then this is quite acceptable for most describing tasks. In the indicated sources 57 and 64, it isn't possible to find a justification for this conclusion.

Reviewer 3 Report

The manuscript in title of "Role of Blockchain Technology in Combating COVID-19 Crisis" is an interesting topic for investigation. For the paper to be accepted, certain refinements need to be made:

  1. The authors must follow the manuscript format of the journal, special in the section of references.
  2. The literature review is too simple to reach the standard of the Applied Sciences journal. Literature analysis needs to be improved.
  3. What are the theoretical and management implications in this study?
  4. In conclusion systematize the advantages and limitations of your research study.
  5. The reviewer recommends the following articles to be cite in this manuscript:
    1.  Fusco, A.; Dicuonzo, G.; Dell’Atti, V.; Tatullo, M. Blockchain in Healthcare: Insights on COVID-19. Int. J. Environ. Res. Public Health 202017, 7167.
    2. Liao, C.-H. Evaluating the Social Marketing Success Criteria in Health Promotion: A F-DEMATEL Approach. Int. J. Environ. Res. Public Health 202017, 6317.
    3. Shen, B.; Guo, J.; Yang, Y. MedChain: Efficient Healthcare Data Sharing via Blockchain. Appl. Sci. 20199, 1207.

Round 2

Reviewer 1 Report

The changes that the authors introduced to address my comments on the previous submission are minimal and inadequate, specially the following  comments 

1- Although a number of challenges have been identified, some of them are presented as general challenges of blockchain technology and they need to be addressed in the context of COVID-19.

2- The proposed solutions of the identified challenges need to be revised as some of them are merely concise recommendations rather than practical solutions.

Reviewer 2 Report

The authors corrected all flaws and answered the issues posed. Now the manuscript increased scientific soundness, many recent projects are described in detail. This is a review that summed up some results, but I missed the original conclusions a little.

Reviewer 3 Report

All comments from the reviewer have been addressed carefully and sufficiently. The revisions are rational from my point of view. The reviewer suggest that the current version of the paper can be accepted.

Author Response

Thank you very much for your efforts and comments

Round 3

Reviewer 1 Report

Accept in present form

Author Response

Thank you for your effort